# Ferritin Metabolism Reflects Multiple Myeloma Microenvironment and Predicts Patient Outcome

**DOI:** 10.3390/ijms24108852

**Published:** 2023-05-16

**Authors:** Federica Plano, Emilia Gigliotta, Anna Maria Corsale, Mojtaba Shekarkar Azgomi, Carlotta Santonocito, Manuela Ingrascì, Laura Di Carlo, Antonino Elia Augello, Maria Speciale, Candida Vullo, Cristina Rotolo, Giulia Maria Camarda, Nadia Caccamo, Serena Meraviglia, Francesco Dieli, Sergio Siragusa, Cirino Botta

**Affiliations:** 1Department of Health Promotion, Mother and Child Care, Internal Medicine and Medical Specialties, University of Palermo, 90127 Palermo, Italycarlotta.santonocito@community.unipa.it (C.S.);; 2Department of Biomedicine, Neurosciences and Advanced Diagnosis, University of Palermo, 90127 Palermo, Italy

**Keywords:** multiple myeloma, smoldering myeloma, monoclonal gammopathy of undetermined significance, ferritin, bone marrow microenvironment

## Abstract

Multiple myeloma (MM) is a hematologic malignancy with a multistep evolutionary pattern, in which the pro-inflammatory and immunosuppressive microenvironment and genomic instability drive tumor evolution. MM microenvironment is rich in iron, released by pro-inflammatory cells from ferritin macromolecules, which contributes to ROS production and cellular damage. In this study, we showed that ferritin increases from indolent to active gammopathies and that patients with low serum ferritin had longer first line PFS (42.6 vs. 20.7 months and, *p* = 0.047, respectively) and OS (NR vs. 75.1 months and *p* = 0.029, respectively). Moreover, ferritin levels correlated with systemic inflammation markers and with the presence of a specific bone marrow cell microenvironment (including increased MM cell infiltration). Finally, we verified by bioinformatic approaches in large transcriptomic and single cell datasets that a gene expression signature associated with ferritin biosynthesis correlated with worse outcome, MM cell proliferation, and specific immune cell profiles. Overall, we provide evidence of the role of ferritin as a predictive/prognostic factor in MM, setting the stage for future translational studies investigating ferritin and iron chelation as new targets for improving MM patient outcome.

## 1. Introduction

Multiple myeloma (MM) is a plasma cell dyscrasia characterized by the proliferation of malignant plasma cells in the bone marrow (BM). MM is a multi-step disease evolving from a pre-malignant condition—monoclonal gammopathy of uncertain significance (MGUS)—to the active disease with an intermediate, asymptomatic form called smoldering myeloma (SMM). Interestingly, while carrying most of the molecular features of active disease, both MGUS and SMM lack clinical signs, suggesting that these events could not be sufficient to justify disease evolution. Along this line, taking into account the strict interplay existing between MM cells and BM cells [1], current studies are trying to understand the major determinants within the microenvironment, contributing to MM pathogenesis [2]. Genomic instability, among others, has the potential to foster MM evolution and BM microenvironment, and is currently being actively studied for therapeutic purposes [3].

Indeed, due to intense replicative and oxidative stress, neoplastic plasma cells generate high amounts of intracellular reactive oxygen species (ROS) leading to DNA damage and genomic instability. Among others, free iron is majorly responsible for ROS mediated damage and cytotoxicity due to its role in initiating pro-oxidant damage. Iron is usually bound to ferritin: an intracellular protein that stores and releases iron in a controlled way and keeps it in a soluble and non-toxic form. In fact, ferritin makes iron available only for critical cellular processes, thus protecting lipids, DNA and proteins from the potentially toxic effects of iron. The ability to accept/release electrons explains the propensity of iron to damage cell components: in excess, iron can generate toxic free radicals via the Fenton reaction where in the presence of hydrogen peroxide, ferrous iron (Fe2+) is oxidized to ferric (Fe3+), producing a hydroxyl radical and hydroxide ion that directly damage cells [4,5].

In the past few years, there has been increased attention to the role of ferritin as a marker of iron overload, reflecting a state of oxidative stress and the pro-inflammatory role of iron in carcinogenesis. Furthermore, ferritin can also be produced by the tumor microenvironment in solid neoplasms. Along this line, several studies also showed that higher serum ferritin levels are associated with worse outcomes in patients with hematologic malignancies [6,7]. This growing interest has led to the study of ferritin in multiple myeloma (MM) and hypothesizing its role as a prognostic marker. Indeed, it has been demonstrated that MM patients with elevated ferritin levels had more adverse prognostic features after induction treatment, such as elevated β2-microglobulin (β2M) and creatinine levels, and low hemoglobin levels. Importantly, patients with relapsed MM displayed increased serum ferritin levels, suggesting its potential role in monitoring disease recurrence [6,8].

In this work we investigated the potential role of ferritin as a prognostic marker in newly diagnosed MM patients (NDMM) and its association with changes in laboratory parameters and immunological BM subpopulations. Finally, we evaluated the impact of gene expression signature associated with ferritin biosynthesis on MM patients outcome.

## 2. Results

### 2.1. Patients Characteristics

A total of 102 newly diagnosed MM patients with available levels of serum ferritin at the time of diagnosis were included in the final analysis. The median ferritin level was 336 ng/mL, so we divided the patients into low (<336 ng/mL) and high ferritin (>336 ng/mL) groups. The median age of patients was 69.66 y (range 36.58–88.95), 46% were female and 54% were male; 16% presented with ISS prognostic stage I, 28% with ISS II, and 47% with ISS III. The main laboratory characteristics of patients are summarized in Table 1. To further support the relevance of ferritin evaluation in MM patients, we performed an additional analysis on a group of MGUS (n.15) and SMM patients (n.17) observed in the same timeframe with available ferritin values at diagnosis (main characteristics reported in Appendix A). Interestingly, as reported in Figure 1A, patients with MM presented a significant highest mean value of serum ferritin (454 ng/mL) as compared to MGUS (139 ng/mL) and SMM (116 ng/mL) patients (*p* < 0.001).

### 2.2. Ferritin Affects MM Patients’ Survival

Subsequently, we evaluated the association of patients’ outcomes (progression free survival (PFS) and overall survival (OS)) with serum ferritin values according to previously described subgrouping (patients with high (HF) and with low blood ferritin (LF)). Interestingly, the median PFS was 20.7 months in the HF group and 42.6 months in the LF group (*p* = 0.047) (Figure 1B). The median OS was 75.1 months for HF vs. NR in patients with low ferritin values (*p* = 0.029).

We then split our database into two subgroups based on eligibility for ASCT. Interestingly, we found that ferritin levels affected the outcomes of the transplant ineligible group only. Specifically, ASCT-ineligible patients with low ferritin had higher median PFS and OS than those with high ferritin (PFS 34.6 vs. 16.3, *p* = 0.027 and OS NR vs. 32.5, *p* = 0.02, respectively) (Appendix A).

### 2.3. High Ferritin Levels Identify a Subgroup of Systemic Inflammatory MM

Next, we examined the correlation between ferritin status and laboratory values extracted from medical records. We found that patients with high ferritin levels had lower hemoglobin values (*p* = 0.001) (Appendix A), as confirmed by a significant negative correlation between ferritin and hemoglobin (Figure 1C). Moreover, patients with high ferritin had higher levels of creatinine (*p* = 0.002), PCR (*p* = 0.03), β2M (*p* = 0.01), NLR (*p* = 0.02) and MLR (*p* = 0.006), and a statistical “trend” was also observed for high monocyte count and the HF group (*p* = 0.07) (Appendix A). These findings suggest a systemic pro-inflammatory status in patients in the HF group. Furthermore, by performing a principal component analysis with laboratory variables for all MGUS, SMM and MM patients, we saw that LF patients clustered closer to MGUS and SMM patients, supporting the idea that patients with a non-inflammatory status could have a less aggressive disease (Appendix A). Additionally, we discovered that patients with osteolytic bone lesions at diagnosis had higher ferritin levels (median value = 393 ng/mL) than those without lesions (median value = 272 ng/mL), *p* = 0.016 (Figure 1D).

### 2.4. FlowCT Analysis: The Bone Marrow Microenvironment

In a subgroup of patients where BM aspirates were available (n: 25, 14 LF and 11 HF), by using FlowCT applied to both PCD and PCST tubes (see Materials and Methods), we investigated the differences in the composition of the bone marrow immune microenvironment. Firstly, by performing a macro-categories analysis (PCD tube) (Figure 2A and Appendix A), we observed that the LF group had a higher frequency of granulocytes (*p* = 0.042) than those in the HF group, in which plasma cells predominated (*p* = 0.016) (Figure 2B). These results underscore the increase in MM cells burden in the HF group, in line with the increased levels of serum B2M previously detected. The subsequent sub-clustering performed with this antibody panel (Figure 2C) failed to identify any additional population differentially represented in the two groups (Appendix A).

Next, we applied a similar workflow to the PCST tube (Figure 2D–F and Appendix A). The sub-clustering of lymphocyte populations allowed us to identify 3 NK cell subsets based on CD38 and CD56 expression. When we compared the abundance of each population between LF and HF groups, we observed a statistically significant increase of CD38^dim^ NK cells in MM patients belonging to the LF group compared to those with high ferritin levels (*p* = 0.038) (Figure 2F). Overall, these results demonstrated that HF MM patients presented an increase in systemic inflammatory markers, as well as in BM PCs, while other normal subpopulations, such as neutrophils and CD38^dim^ NK, are significantly reduced.

### 2.5. Identification of Ferritin Related Genes in MM Patients Using a Single-Cell-like Approach

To gain further insight into the role of ferritin in MM patients, we performed a bioinformatic analysis on the CoMMpass database, focusing on genes involved in ferritin production/biosynthesis (as reported in material and methods). To do that, a total of 859 transcriptomes (one for each patient at baseline) were included (Figure 3A). After quality control, these data were analyzed by using a “single-cell-like” approach: we first used dimensional reduction (PCA and UMAP), and next a clustering based on a ferritin related gene set that led us to identify six different patient subclusters (Figure 3A, main genes expression are reported below the clustering UMAP) (Appendix A). Both ASCT eligible (n = 420) and not-eligible (n = 439) patients were included in our analysis. The gene set enrichment analysis on Cluster (C) 3, 4 and 5 showed a significant enrichment score for the ferritin related gene set (padj = 0.002, ES = 0.8) (Figure 3B) with a significant enrichment of FTL and FTH1 (adjp < 0.001): accordingly, patients belonging to these clusters were identified as the high ferritin group (HF). Interestingly, among the 3 ferritin-enriched clusters, C5 revealed a highly proliferative potential (an excess of genes belonging to S phase was found) (Figure 3A).

Once evaluated for survival outcome, we observed that, despite being ASCT eligible or not-eligible, patients belonging to the HF group presented a worse OS and PFS. Among them, patients belonging to the C5 cluster presented overall worse survival (Appendix A).

### 2.6. Single Cell (sc) RNAseq Data Confirm Results from Flow Cytometry and Bulk Transcriptomic Analysis

To validate our results further, we integrated sc-RNAseq data from 3 different datasets with MM cells and immune cells from 19 MGUS, 10 SMM and 17 MM patients (Figure 4A and Appendix A). First, we verified that FHT1 and FTL genes increased significantly from MGUS to MM patients (Appendix A). Then, we correlated the distribution of the different immune subpopulations, as identified by using the map to reference function in the *Seurat* package with the presence of high or low levels of FTH1 and FTL. As shown in Figure 4B, we confirmed the decrease of NK cells (as seen in the flow cytometry analysis), the increase of monocytes (as reported by laboratory variables), and the decrease of B naïve cells (as observed as a trend in the PCD tube analysis) in patients with high FTH1 or FTL1 levels. Moreover, we detected other populations that were differentially represented in the subgroups (Appendix A), and among them, we noticed a reduction of CD4 naïve cells in patients with high levels of FTH1/FTL.

## 3. Discussion

Iron is fundamental for many cellular functions and its metabolism is critical for the maintenance of homeostasis in humans. Ferritin represents a marker of iron stores and is notably elevated in the presence of inflammatory conditions. Along the same line, in hyperinflamed tumors [9], the associated microenvironment becomes iron-enriched due to its release from pro-inflammatory cells (including tumor-infiltrating macrophages and granulocytes). However, the excess of iron ultimately promotes cancer cell progression by inducing immune suppression through T (and B) lymphocytes anergy and apoptosis [10], and by impairing antigen-presenting cells (APCs) activity [11]. Moreover, increased iron availability could enhance the rate of cell replication; in fact, it has recently been demonstrated that high ferritin levels are associated with a worse prognosis in breast cancer [12,13].

Some studies have proposed ferritin as a negative prognostic factor in MM, as increased levels were shown to correlate with stage III disease and higher values of B2M, IL-6 and LDH expression [14]. These data are consistent with our results, where we showed an increase in mean ferritin values from MGUS to MM conditions and a significant rise of B2M in the HF group of MM patients.

In addition, ferritin production is also induced by proinflammatory cytokines and up-regulated by NF-kB, which is often overactivated in MM [15,16]. We indeed observed in HF patients an increase in serum markers of systemic inflammation (ESR/CRP), as well as an increased probability to develop osteolytic lesions.

Furthermore, higher ferritin levels inhibit tumor necrosis factor-induced apoptosis; thus, supporting myeloma cell survival [7]. Accordingly, an Austrian study showed that pre-transplant high ferritin levels worked as an independent, negative prognostic factor for PFS and OS in MM patients eligible for ASCT [15]. Although in our series this result was confirmed for transplant-ineligible patients only (possibly due to the low number of transplant eligible patients), our bioinformatic approach fully supported these findings by further showing a specific enrichment of S-phase genes in a subgroup of ferritin enriched patients.

To the best of our knowledge, in our study, we performed, for the first time, an attempt to correlate bone marrow microenvironment composition with serum ferritin, to study specific correlations with both cancer and immune cells. Our findings indirectly support the role of ferritin in promoting MM cell proliferation: indeed MM patients belonging to the HF group presented a significant increase in the number of BM plasma cells. On the other hand, we found a lower NK CD38^dim^ number in the same group of patients. This specific finding is of utmost importance if we consider the role of these cells against tumors and in antibody-dependent cytotoxicity [17,18]. Finally, by performing an integrative bioinformatic analysis on three available scRNAseq datasets with plasma cells and immune cells from the bone marrow of MGUS, SMM and MM patients, we fully confirmed the association between high levels of ferritin correlated genes and changes in the immune microenvironment (including reduced NKs and increased monocytes).

Our data provide evidence of the importance of ferritin evaluation in the work-up of MM patients’ assessment and follow-up. Therefore, we identified ferritin as a potential new target for MM treatment, laying the groundwork for future combinatory studies that should include iron chelation as a backbone to enhance immunomodulatory agents or mAbs. Some preclinical evidence already supports this point [7,14,19,20]. Finally, further evidence is needed to fully validate ferritin as a new prognostic factor for MM, and to determine its role as a new potential target to improve patient outcome.

## 4. Materials and Methods

### 4.1. Patient Population

This is a single-center retrospective study (approved by our internal ethical committee with the number 02/2022, codename: MMVision) involving 102 NDMM, 17 smoldering MM (SMM) and 15 monoclonal gammopathy of undetermined significance (MGUS) patients treated at our institution, “Paolo Giaccone” University Hospital between January 2019 and May 2022, for whom ferritin was available at the time of diagnosis of MM, according to the International Myeloma Working Group (IMWG) criteria. Patients were excluded from the analysis if they: (1) did not sign the informed consent; and (2) bear any other pathology or medical condition that in the opinion of the investigator may interfere with adherence to the protocol or with the expression of informed consent.

### 4.2. Data Collection

Baseline demographic and clinical data were retrospectively collected from medical records. The following indicators were retrieved: (1) demographic characteristics, including age and baseline diseases; (2) type of MM and staging; (3) laboratory data, including white blood cells, hemoglobin, platelets, serum creatinine, albumin, globulin, calcium, lactate dehydrogenase, ferritin, β2M, neutrophil/lymphocyte ratio, and monocyte/lymphocyte ratio; and (4) median PFS and OS.

### 4.3. Statistical Analysis

Multivariate-adjusted Cox regression and K m curves were used to analyze the association of high levels of serum ferritin with PFS/OS in patients with MM.

Continuous data are shown as mean ± standard deviation or median. Categorical variables are presented as numbers or percentages. Differences between the two groups were assessed using the Student’s *t*-test, chi-square test, or Mann–Whitney U test. Two-tailed statistical analysis was used, and *p*-values of <0.05 were considered statistically significant.

### 4.4. BM Aspirates Preparation

A total of 25 (out of 102) MM patients (n = 14 with low ferritin levels and n = 11 with high ferritin levels) presented BM available for further characterization and were enrolled for the flow cytometric immunophenotyping of bone marrow microenvironment using the BD OneFlow™ PCD and Plasma Cell Screening Tube (PCST) (Beckton Dickinson, Franklin Lakes, NJ, USA) containing the following fluorochrome-conjugated antibodies: anti-human CD38 FITC, anti-human CD28 PE, anti-human CD27 PerCP Cy5.5, anti-human CD19 PE-Cy7, anti-human CD117 APC, anti-human CD81 APC-H7, anti-human CD45 BD Horizon V450, anti-human CD138 BD Horizon V500-C, anti-human CD56 PE, anti-human b2-microglobulin PerCP Cy5.5, anti-human Kappa APC, and anti-human Lambda APC-H7 antibodies.

BM aspirates were collected in EDTA tubes and stained according to the producers’ guidelines. Once washed, samples were acquired on the BD FACSLyric™ Flow Cytometer (Beckton Dickinson, Franklin Lakes, NJ, USA).

### 4.5. Dimensionality Reduction and FlowCT Analysis

Subsequently, we applied FlowCT [21], an R-based package, to identify possible differences in the immune microenvironment among multiple myeloma patients with low or high ferritin levels. FlowSOM and Seurat clustering approaches were used for automated clustering. The median expression of each marker on multi-uniform manifold approximation and projection (UMAP) and the Infinicyt software (Infinicyt v2.0; Cytognos SL, Salamanca, Spain) were used to characterize each cluster.

### 4.6. Gene Set Analysis

We analyzed Bulk RNA-seq from CD138-positive fraction of BMMC from a total of 859 samples coupled with clinical data that were provided by the Multiple Myeloma Research Foundation (MMRF) CoMMpass study (NCT01454297). Patients were grouped according to Autologous stem cell transplantation (ASCT) eligibility: ASCT eligible = 420, median age = 60 not-eligible for ASCT= 439 and, median age = 67. Firstly, we developed a ferritin related gene-set including 49 genes by merging gene ontology and Molecular Signatures Database (MSigDB). Lastly, enriched GO terms from the biological processes and KEGG pathways for the selected genes (ferritin biosynthesis) was obtained using ClueGO plug-in of Cytoscape v3.9.1 [22].

These gene set libraries were used as the backbone for all subsequent analysis. RNA-seq data of all MM subjects were pre-processed, log2-transformed, and analyzed using the DEseq2 R package. biomaRt (v2.48.2) was used for annotation based on GRCh37/hg19 to cross-map gene symbol identification. The final count matrix contained 19282 FPKM and was projected in the Seurat object, where after feature selection and scaling of the normalized data, we performed PCA linear dimensional reduction. The first 20 PCs were used to construct the KNN graphic, and the Louvain algorithm was performed for clustering the patients with a resolution parameter set to 0.5. We ran the UMAP method for the non-linear dimensional reduction to visualize the entire dataset.

Cell cycle phases were scored using the list of cell cycle markers from Tirosh et al. that are preloaded in Seurat [23]. Gene set enrichment analysis (GSEA) was performed using the FGSEA R package (v1.25.1) to determine whether scRNA-seq data from bulk GEP that can be used to classify individual patients based on the developed Ferritin gene-set. The final patients clustering was used to gene set enrichment to the Kaplan–Meier (KM) survival analysis and to discover the relationships between clinical features and ferritin related genes alteration.

### 4.7. Single Cell Analysis

Data from MGUS, SMM and MM patients were retrieved from GSE145977 [24] GSE124310 [25] and GSE163278 [26] datasets. scRNAseq results were extracted and merged together using the R package Seurat. Batch removal has been performed through the Harmony package. FTL and FTH1 genes where retrieved from each patient and the mean value among all cells was used as patient value. Patients were dichotomized according to FTH1 and FTL1 median values and a double analysis has been performed to identify associations between high and low levels of either FTL or FTH1 mRNA and immune cells composition. Immune cell annotations have been performed according to Seurat guidelines by evaluating cell projection to a reference dataset.

## Figures and Tables

**Figure 1 ijms-24-08852-f001:**
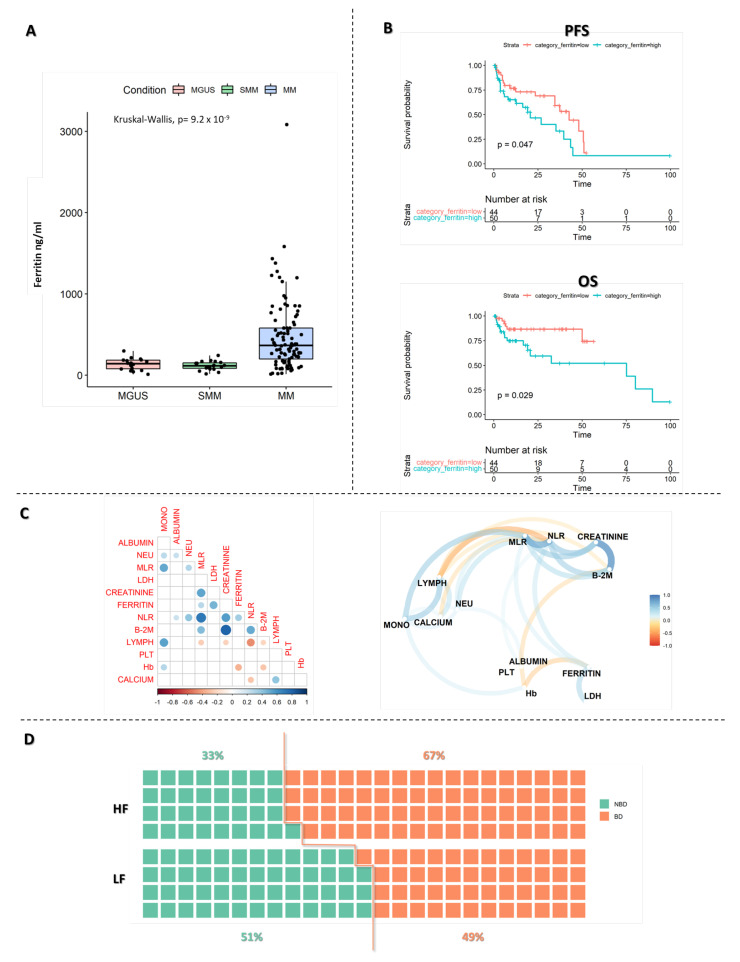
Survival probability and multivariate analysis. (**A**) Ferritin values in MGUS, SMM and MM patients. (**B**) Progression free survival (PFS) and overall survival (OS) in patients with high ferritin and those with low ferritin. (**C**) Analysis of direct correlations between biochemical variables and network analysis of relationships. (**D**) Difference in percentage incidence of bone disease (BD) between subjects with high ferritin levels and those with low levels.

**Figure 2 ijms-24-08852-f002:**
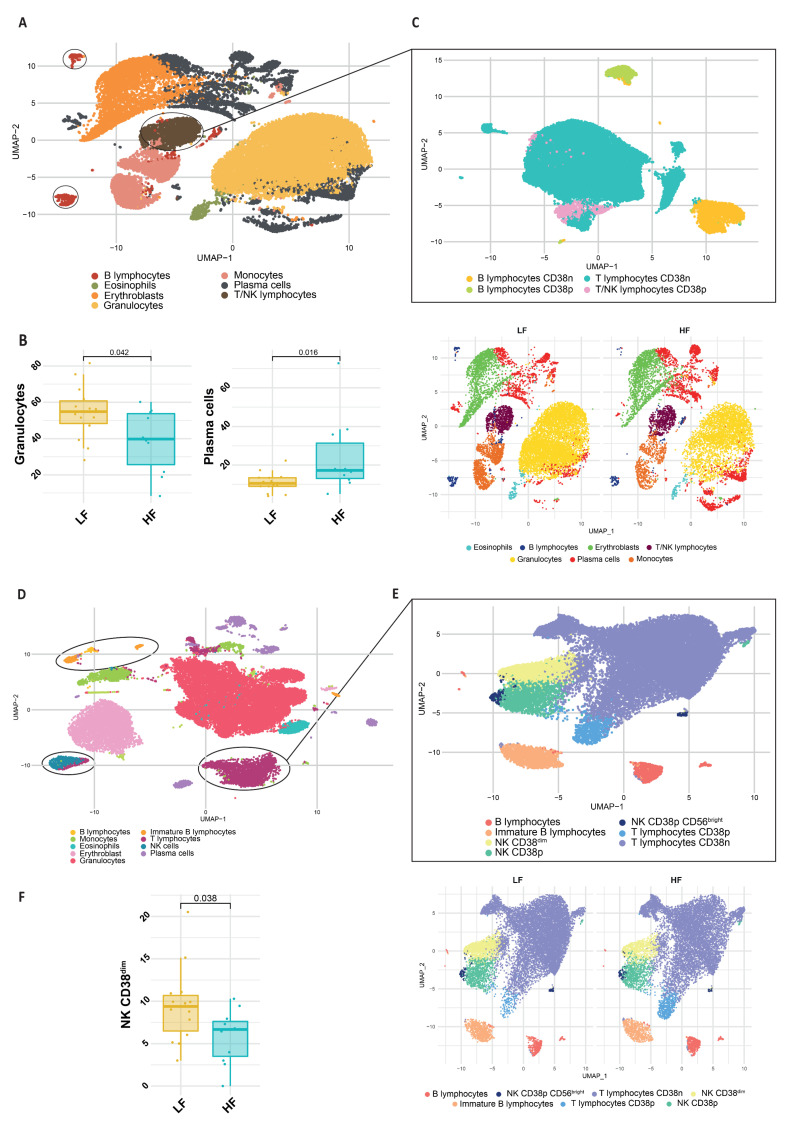
FlowCT analysis of BD OneFlow™ PCD and PCST tubes. (**A**) Uniform manifold approximation and projection (UMAP) of macropopulations (eosinophils, erythroblasts, granulocytes, B and T/NK lymphocytes, monocytes and plasma cells) identified in the PCD tube. (**B**) Boxplots and UMAPs show the different distribution of these populations between low ferritin (LF) and high ferritin levels group (HF) (*p* value < 0.05). (**C**) Subclustering focusing on lymphocyte subsets. (**D**) UMAP of macropopulations (eosinophils, erythroblasts, granulocytes, B and T lymphocytes, immature B lymphocytes, monocytes, NK cells, and plasma cells) identified in the PCST tube. (**E**) Subclustering of NK cells, B and T lymphocytes. (**F**) Boxplots and UMAPs showing the different distribution of NK cells and B/T lymphocytes between LF and HF groups (*p* value ≤ 0.05).

**Figure 3 ijms-24-08852-f003:**
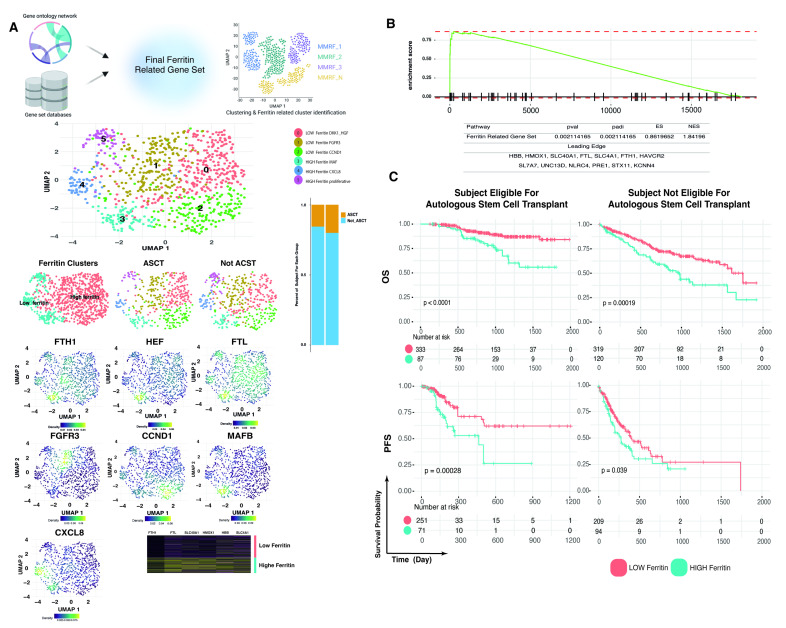
Ferritin related clustering on PCs derived from MM patients at single cell resolution. (**A**) Workflow of the whole analysis. Below UMAP, reporting patients clustering and main genes expressed in each cluster. (**B**) GSEA on high ferritin versus low ferritin group show a significant enrichment score (ES) for the high ferritin cluster. (**C**) Kaplan–Meier curves of ferritin-enriched clusters versus non-enriched clusters based on ASCT eligibility reporting overall survival (OS) and progression free survival (FPS).

**Figure 4 ijms-24-08852-f004:**
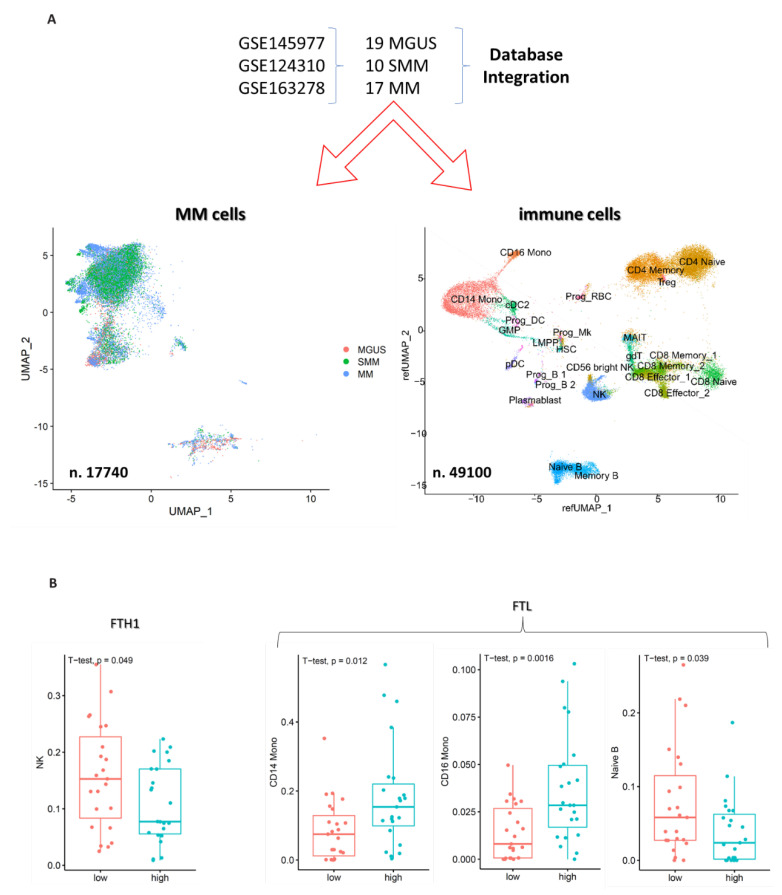
Single cell analysis of MM and immune cells. (**A**) Workflow of the integrative analysis. Below UMAPs, reporting plasma cells clustering according to disease status (left) and immune cells clustering including cell annotations (right). (**B**) Boxplot reporting main differences in cell populations distribution confirming what was observed through flow cytometry and laboratory data.

**Table 1 ijms-24-08852-t001:** The most important laboratory features of enrolled MM patients.

	Median	Range
Hb (g/dL)	9.90	6.40–17
Neutrophils (µL)	3320	900–18,890
Lymphocytes (µL)	1840	170–6860
Monocytes (µL)	535	40–780
Platelets (µL)	217,000	285–264,000
Creatinin (mg/dL)	1.12	0.05–8.19
Albumin (g/L)	3.43	1.89–34
Calcium (mg/dL)	9.35	7.02–14.40
β2M (mg/dL)	5.12	0.40–52
LDH (U/L)	182	43–1045
ESR (mm/h)	46	3–120
CRP (mg/L)	4.54	0.07–140
CM (g/dL)	3.09	0.02–37
k chains (g/L)	1.77	0.02–28.93
λ chains (g/L)	1.35	0.03–24.31
Ferritin (ng/mL)	336	11–3084

## Data Availability

Not applicable.

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
