# Peer review of "Ferritin Metabolism Reflects Multiple Myeloma Microenvironment and Predicts Patient Outcome"

_ijms, 2023, doi:10.3390/ijms24108852_

Round 1
Reviewer 1 Report
In this manuscript, entitled, “Ferritin metabolism reflects multiple myeloma microenvironment and predicts patient outcome” by Plano et al., the authors propose the role of ferritin as a prognostic marker for multiple myeloma. The role of iron and ferritin in multiple myeloma patients has been well characterized. The functional implication of the increased ferritin levels on the immune microenvironment is interesting and timely. However, the study seems very premature and further analysis and validation is required.
Major concerns:
1. The aspect of differential immune infiltration in ferritin high and low cohort is interesting. The authors must utilize available transcriptomic datasets of MM patients and perform detailed analyses using alternate databases/software to validate the findings.
2. The authors have confirmed the previous findings using their patient cohort. Can the authors include laboratory features of additional normal control patient cohorts?
Minor concerns:
1. Figure 1D-the legend is missing.
Author Response
We thank the Reviewer for these positive comments, we tried to improve the manuscript according to suggestions received.
Major concerns:
- The aspect of differential immune infiltration in ferritin high and low cohort is interesting. The authors must utilize available transcriptomic datasets of MM patients and perform detailed analyses using alternate databases/software to validate the findings.
- We thank the reviewer for this suggestion that clearly improved the quality of the manuscript. We included single cell transcriptomic data available from three different studies. We normalized and aligned the samples and this approach gave us the possibility to analyze at the same time changes in the immune microenvironment correlated to the expression of ferritin related genes in MM associated neoplastic cells. We thus confirmed most of the main results we obtained trough flow cytometry and laboratory analyses.
- The authors have confirmed the previous findings using their patient cohort. Can the authors include laboratory features of additional normal control patient cohorts?
- We believe that this is a crucial point. Unfortunately, our study was born and approved as a single center institution and it is very difficult for us to receive data from other center at this point. By the way, to mitigate the bias potentially introduced by a MM (mono)-centric approach, we analyzed the ferritin (and other lab parameters) values obtained from MGUS and SMM patients. These results, confirming the strong raise in ferritin levels moving from MGUS to MM are currently reported in the first paragraph of the results section. Additionally, we added data from 3 different single cell datasets, all confirming pour results.
Minor concerns:
- Figure 1D-the legend is missing.
- We really apologize for the error. We have included it in the revised version of the manuscript
Reviewer 2 Report
In this study, the researchers suggested that the ferritin level in the serum can be a predictive/prognostic factor in multiple myeloma (MM) patients. They found that patients with lower serum ferritin level have better survival rate both in progression free survival (PFS) and overall survival (OS). To explore the potential mechanism of ferritin as a prognostic factor, they performed FlowCT analysis to investigate the bone marrow microenvironment between patients with high ferritin (HF) and low ferritin (LF). The results show that HF patients have more inflammatory microenvironments compared to LF patients. By analyzing transcriptome data in CoMMpass database, the authors found some gene cluster are enriched in ferritin metabolic pathway, and patients with high ferritin level in CoMMpass database also have worse OS and PFS. These results are preliminary, however, they suggested proof-of-principle of the role of serum ferritin as a prognostic marker in MM. Before its publication, there are some aspects can be improved, see the comments below:
1. Line 211, 212; the author state that “patients with 210 higher ferritin presented increased creatinine levels ….” there is no figure or dataset to support this sentence, please provide them.
2. Figure 2A, is this figure present the results from HF patients PCD tube or LF patients?
3.Figure 2B, what’s the unit of y-axis, cell number or the percentage, please clarify.
4. Line 220, there is a typo “wer” should be “were”.
Author Response
We thank the Reviewer for this elegant resume and for the comments, we tried to improve the manuscript according to suggestions.
- Line 211, 212; the author state that “patients with 210 higher ferritin presented increased creatinine levels ….” there is no figure or dataset to support this sentence, please provide them.
We are very sorry for this mistake. The boxplots reporting this and other results was erroneously not reported in the previous version of the manuscript. We corrected this point.
- Figure 2A, is this figure present the results from HF patients PCD tube or LF patients?
Thanks to the reviewer for the request. Figure 2A represented the UMAP obtained from PCD tube of all patients without classifying them according to ferritin levels, which were shown in Figure 2B. In Figures 2D and 2F, the same analysis was performed on the PCST tube.
3.Figure 2B, what’s the unit of y-axis, cell number or the percentage, please clarify.
Thanks to the reviewer for the request. The box plots in Figure 2B showed the percentages of granulocytes and plasma cells between HF and LF patient groups, as in Figure 2F for NK CD38dim.
- Line 220, there is a typo “wer” should be “were”.
We apologize for this typos. We have changed the word to the correct one.
Round 2
Reviewer 1 Report
The authors have satisfactorily addressed my concerns. The manuscript may be accepted in the present form for publication after moderate English language corrections.